# Subgroup fairness in two-sided markets

**Quan Zhou**[1,2]*, **Jakub Mareček**[3], **Robert Shorten**[1,2]

**1** Dyson School of Design Engineering, Imperial College London, London, United Kingdom, **2** School of Electric and Electronic Engineering, University College Dublin, Dublin, Ireland, **3** Department of Computer Science, Czech Technical University in Prague, Prague, Czech Republic

* q.zhou22@imperial.ac.uk

**Data Availability Statement:** All relevant data are within the paper and its Supporting information files.

**Funding:** The work of Quan Zhou and Rober Shorten has been supported by the Science Foundation Ireland under 379 Grant 16/IA/4610.

## Abstract

It is well known that two-sided markets are unfair in a number of ways. For example, female drivers on ride-hailing platforms earn less than their male colleagues per mile driven. Similar observations have been made for other minority subgroups in other two-sided markets. Here, we suggest a novel market-clearing mechanism for two-sided markets, which promotes equalization of the pay per hour worked across multiple subgroups, as well as within each subgroup. In the process, we introduce a novel notion of subgroup fairness (which we call Inter-fairness), which can be combined with other notions of fairness within each subgroup (called Intra-fairness), and the utility for the customers (Customer-Care) in the objective of the market-clearing problem. Although the novel non-linear terms in the objective complicate market clearing by making the problem non-convex, we show that a certain non-convex augmented Lagrangian relaxation can be approximated to any precision in time polynomial in the number of market participants using semidefinite programming, thanks to its "hidden convexity". This makes it possible to implement the market-clearing mechanism efficiently. On the example of driver-ride assignment in an *Uber*-like system, we demonstrate the efficacy and scalability of the approach and trade-offs between Inter- and Intra-fairness.

## 1 Introduction

In two-sided markets [1], the exchange between two distinct user groups is intermediated by a platform that matches supply to demand. As two-sided markets for services of the so-called "gig economy" grow, fairness thereof is becoming important both from an ethical and legal point of view [2–4]. For example, the former chief economist of Uber [5] documented a roughly 7% gender earnings gap among drivers, but contented that the gender earnings gap might be due to men choosing their time and location of rides better, and also due to men accumulating experience quickly [5, 6], considering that men can work more hours per week than women in the study. Indeed, many other studies [7–9] show that women are much less likely than men to work regularly in two-sided markets. It was recently reported that in the US, gig work constituted a lower share of total earnings for women than for men: [7] report 16%, versus 23% for men. In the UK, [8] found that 75% of female gig workers earned less than £11,500 per annum, compared with 61% of all workers. Although these issues have been

Jakub Mareček acknowledges support of the OP RDE funded project 380 CZ.02.1.01/0.0/0.0/16_019/0000765 "Research Center for Informatics". This work has received funding from the European Union's Horizon Europe research and innovation programme under grant agreement No. 101070568. This work was also supported by Innovate UK under the Horizon Europe Guarantee; UKRI Reference Number: 10040569 (Human-Compatible Artificial Intelligence with Guarantees (AutoFair)).

**Competing interests:** The authors have declared that no competing interests exist.

longstanding in labour-market structures, the COVID-19 pandemic has set women's labour force participation back more than 30 years, according a report from U.S. Department of Labour [10] and exposed the urgent need for better solutions that support all working women. We aim to develop market-clearing mechanisms for two-sided markets that would explicitly improve subgroup fairness, while being efficiently computable.

In the context of a two-sided market, where workers are matched with jobs by a platform, we define efforts to provide better services to both sides as Worker-Care and Customer-Care, respectively. From the perspective of workers, we suggest that one should distinguish between the following notions of fairness:

1. Intra-fairness: Motivated by the concepts of proportional fairness in [11], the notion of Intra-fairness pursues the proportional parity of worker utility (e.g., salary and allowance). It requires that over time, the accumulated utility, proportional to the workload (e.g, working hours and active time) for every worker, should be equalised.

2. Inter-fairness: By analogy with demographic parity, our notion of Inter-fairness strives to ensure that the proportion of accumulated utility to the workload should be equalized across the (majority and minority) subgroups. A typical example of Inter-fairness would be closing up the gender gap of accumulated utility, proportional to the workload. In the context of this notion of fairness, overlapping subgroups [12] can also be accommodated.

While there have been some attempts to equalize pay and opportunity across all workers [11, 13, 14] (Intra-fairness, to use our terms), the notion of subgroup fairness (or Inter-fairness) has received very little attention. This may ignore significant gaps across different subgroups, e.g., gender-earning gaps, because the algorithm could approach Intra-fairness by minimizing the worker utility within subgroups. In contrast, our goal is to equalize the pay per hour worked both across subgroups based on gender, ethnicity, or other sensitive attributes, as required by current or proposed [15] legislation, and within each subgroup.

Notice that the notion of Intra-fairness might be in conflict with Inter-fairness, but that it may be possible to improve one of the objectives without worsening the other too much. We present a natural formulation of the market-clearing problem utilizing a weighted sum of these objectives and its augmented Lagrangian relaxation formulation. We also illustrate the trade-off between Intra- and Inter-fairness computationally.

## 2 Related work

The last few years have seen an unprecedented explosion in the attention of fairness [e.g., 12, 13, 16–20] in artificial intelligence and machine learning. Much of the work focuses on classification [16, 17, 19, 21] and recommendation systems [e.g., 14, 22], and specifically on the notions of subgroup fairness therein.

Trivially, one aim is to achieve this by "fairness under unawareness", where the protected attributes are not given. The shortcoming of this approach is obvious when there are some other features related to protected attributes and those features could be used to predict the protected attributes. On the contrary, "demographic parity"requires that the proportion of each segment of a protected class (e.g., gender) should receive the positive outcome at equal rates, although this could be unfair in the case of unbalanced distributions of features between advantaged and disadvantaged subgroups, even in the absence of biases. The notions of "equal opportunity" in [16] and "counterfactual fairness" of [23] require the predictor to be unrelated to protected attributes. All of these notions relate to subgroups of the population and provide an average guarantee for individuals in the protected group [24].

Independently, the individual definition of fairness asks for constraints to equalize the outcome across all pairs of individuals, rather than across subgroups. In other words, it requires that "similar individuals should be treated similarly" [21]. This notion of fairness has been considered by [11, 13] in the context of ride-hailing, in two papers most closely related to ours, albeit with exponential-time solvers for the simpler market-clearing problem. However, this notion requires a similarity metric to capture the truth of the ground, which relies on a task-specific assumption [25]. Considering that multiple conflicting notions of fairness are possible, [26] design an algorithm such that individual fairness is maximized when enforcing group-fairness constraints. We should like to consider these multiple constraints in matching problems. [27, 28], and to provide efficient market-clearing mechanisms. [29] explores the fairness in matching market from the perspective of bandit learning. Another instance is the fairness resolution model [24], guided by the unfairness complaints received by the system. It could be a more practical way to maintain both group and individual fairness. We refer to [30] for a detailed survey of mainstream fairness notions and corresponding methodologies.

On the other hand, there is vast research on specific problems in two-sided markets, ranging from matching between different types of users [31–33], market equilibrium [34], price discrimination issues [34–36], to fair ranking or recommendation systems [37–40]. Especially, considering users' gender preference, [36] propose a "female-only" subsystem that matches safety-concerned female riders to female drivers.

## 3 Modelling fairness in two-sided markets

Two-sided markets [1] model platforms that enable the interaction between the two sides: customers submitting jobs $i \in \mathcal{C}_t$, and workers $j \in \mathcal{D}$ throughout the time window $t \in \mathcal{T}$. For example, the system of Uber can be modelled as a two-sided market with direct connections. In this case, workers would be Uber drivers and jobs could be the rides of customers. Note that jobs are assumed to be independent of each other, although one customer might offer multiple jobs.

### 3.1 Notation

In period $t$, a list of jobs $i \in \mathcal{C}_t$ is offered with payment $\text{pay}_t^i$ for each job to all workers $j \in \mathcal{D}$ in the two-sided market. A binary variable $A_t^j$ indicating all workers' availability in period $t$, i.e., $A_t^j = 1$ if worker $j$ is online and waiting for a job in period $t$. Accordingly, we split all workers $\mathcal{D}$ into two sets: $\mathcal{D}^{A_t=1}$ and $\mathcal{D}^{A_t=0}$, by their availability such that all available workers in period $t$ are in the set $\mathcal{D}^{A_t=1}$ and the unavailable in the set $\mathcal{D}^{A_t=0}$. Further, another binary variable $M_t^{(i,j)}$ is set to be 1 if worker $j$ is assigned job $i$ in period $t$. Note that worker $j$ cannot get any jobs if she is offline and we assume that one worker can get at most one job in a single period, thus $\sum_{i \in \mathcal{C}_t} M_t^{(i,j)} \le A_t^j$, for $j \in \mathcal{D}, t \in \mathcal{T}$. Also, each job must be matched with exactly one worker, such that $\sum_{j \in \mathcal{D}} M_t^{(i,j)} = 1$, for $i \in \mathcal{C}_t, t \in \mathcal{T}$.

Let $d_t^{(i,j)}$ denote the suitability between job $i$ and worker $j$ in period $t$. Specifically, $d_t^{(i,j)}$ can be considered as the distance between the worker $j$ and the pick-up location of the job $i$ in period $t$, or as the (estimated) customer waiting time for pick-up. Note that $d_t^{(i,j)}$ is normally unavailable if the worker is offline. From the customers' point of view, their service experience ("Customer-Utility") is highly related to suitability $d_t^{(i,j)}$, especially, the waiting time of customers. Here, if the job $i$ is accepted by the worker $j$ in period $t$, we set

$$\text{Customer-Utility} := -d_t^{(i,j)}, \tag{1}$$

for $i \in \mathcal{C}_t$. On the other hand, workers need to cover the cost of driving to pick-up location, and a long wait for pick-up would negatively affect their customer reviews. For simplicity, we assume Worker-Utility $j$ is only related to $\text{pay}_t^i$ and $d_t^{(i,j)}$ if she receives the job $i$ in period $t$. But the utility would be 0 if she does not receive any jobs or is offline. We have

$$\text{Worker-Utility} \quad u_t^j := \sum_{i \in \mathcal{C}_t} M_t^{(i,j)} \left( \text{pay}_t^i - d_t^{(i,j)} \right), \tag{2}$$

for $j \in \mathcal{D}^{A_t=1}$ and $u_t^j := 0$, for $j \in \mathcal{D}^{A_t=0}$. In addition, we define $U_t^j$ as the accumulated utility of the worker $j$ at the end of the period $t$, such that $U_t^j = U_{t-1}^j + u_t^j$, for $t \in \mathcal{T} \setminus \{0\}$ and $U_0^j = 0$, for $j \in \mathcal{D}$. Further, we define $\Lambda_t^j$ to be the accumulated workload of worker $j$ at the end of period $t$. For simplicity, $\Lambda_t^j$ is calculated by the number of periods in which worker $j$ is available till period $t$: $\Lambda_t^j = \sum_{k=1}^t A_k^j$, for $j \in \mathcal{D}, t \in \mathcal{T}$. We denote the ratio of accumulated utility proportional to workload ($U_t^j / \Lambda_t^j$) as the return rate of worker $j$ til the end of period $t$.

Please refer to Table 1 for an overview of the major notation. In real life, the worker would rather reject a job if it cannot bring any benefit, i.e., $u_t^j \geq 0$, for $j \in \mathcal{D}$. However, if workers are allow to reject jobs, it might happen that certain jobs could not be matched with any worker. In other words, $u_t^j \geq 0$, together with the constraint $\sum_{j \in \mathcal{D}} M_t^{(i,j)} = 1$, would lead to infeasibility of the optimisation programs. In the following formulations, we would keep constraint $\sum_{j \in \mathcal{D}} M_t^{(i,j)} = 1$ but $u_t^j \geq 0$. Note that the conflict between these two constraints could be easily softened by Lagrangian relaxation.

## 3.2 Fairness notions

In a two-sided market, the platform often makes efforts to provide better services for end-users, although there might be some trade-offs between both sides. We can define the terms Customer-Care and Worker-Care to distinguish the efforts invested on both sides. In the case of ride-sharing platforms, Customer-Care is set directly to be the sum of Customer-Utility, as in (3) and we always expect to maximise Customer-Care.

$$\text{Customer-Care} := -\sum_{i \in \mathcal{C}_i} \sum_{j \in \mathcal{D}^{A_t=1}} M_t^{(i,j)} \times d_t^{(i,j)}, \tag{3}$$

We have a detailed discussion of Worker-Care in the following text.

**Table 1. A brief overview of notation.**

| Symbols | Definitions |
|---|---|
| $\mathcal{C}_t$ | a set of jobs in period $t$. |
| $\mathcal{D}$ | a set of workers. |
| $\mathcal{T}$ | the time window. |
| $\text{pay}_t^i$ | the payment of job $i$. |
| $u_t^j$ | the utility of worker $j$ in period $t$. |
| $U_t^j$ | the accumulated utility of worker $j$ til period $t$. |
| $A_t^j$ | the availability of worker $j$. |
| $\Lambda_t^j$ | the accumulated workload of worker $j$ til period $t$. |
| $d_t^{(i,j)}$ | the suitability between job $i$ and worker $j$. |
| $M_t^{(i,j)}$ | the indicator of assigning job $i$ to worker $j$. |

Intra-fairness: proportional parity of Worker-Utility: One would like to ensure that for all pairs $(j_1, j_2)$ of workers, the difference between the return rates $U_t^{j_1}/\Lambda_t^{j_1}$ and $U_t^{j_2}/\Lambda_t^{j_2}$ is as small as possible. The pairwise differences can be evaluated in a number of ways. A large family of such measures of inequality in a population, due to [41], is known as generalised entropy indices $GE(\alpha), \alpha \in \mathbb{R}$. The earlier indices developed by [42, 43] are special cases of $GE(1)$, $GE(0)$ within this family. Another popular measure of inequality is the Gini index, for which we use the aggregate-value formulation of [44], instead of the formula from the World Bank [45], which requires probability distribution of incomes or ranking of incomes. In addition, we also compare against another, linearised notion of proportional parity, which has been proposed by [11].

In general, we denote the term representing Intra-fairness till period $t \in \mathcal{T}$ by Intra-Fair$_t$, and define it case-wise:

$$
\begin{cases}
\dfrac{1}{\alpha(\alpha-1)|\mathcal{D}|} \sum_{j \in \mathcal{D}} \left( \left( \dfrac{U_t^j}{\Lambda_t^j \times \mu_t} \right)^\alpha - 1 \right) & \text{for } GE(\alpha)_t \\[3mm]
\dfrac{1}{|\mathcal{D}|} \sum_{j \in \mathcal{D}} \dfrac{U_t^j}{\Lambda_t^j \times \mu_t} \ln \dfrac{U_t^j}{\Lambda_t^j \times \mu_t} & \text{for } GE(1)_t \\[3mm]
\dfrac{1}{|\mathcal{D}|} \sum_{j \in \mathcal{D}} \ln \dfrac{\Lambda_t^j \times \mu_t}{U_t^j} & \text{for } GE(0)_t \\[3mm]
\dfrac{1}{2\mu_t |\mathcal{D}|^2} \sum_{j_1, j_2 \in \mathcal{D}} \left| \dfrac{U_t^{j_1}}{\Lambda_t^{j_1}} - \dfrac{U_t^{j_2}}{\Lambda_t^{j_2}} \right| & \text{for Gini}_t \\[3mm]
\sum_{j \in \mathcal{D}} \left| \max_{k \in \mathcal{D}} \dfrac{U_{t-1}^k}{\Lambda_{t-1}^k} - \dfrac{U_t^j}{\Lambda_t^j} \right| & \text{for linearised,}
\end{cases}
\tag{4}
$$

where $\alpha \neq 0, 1$ and $\mu_t := \frac{1}{|\mathcal{D}|} \sum_{j \in \mathcal{D}} U_t^j / \Lambda_t^j$ is the average return rate till period $t$. Lower Intra-fairness indicates higher equality and zero Intra-fairness means absolute equality.

Inter-fairness: gender gap of Worker-Utility: The notion of Inter-fairness requests the proportional parity of Worker-Utility among subgroups. If we pick one worker randomly from each subgroup, the difference among their return rates is as small as possible. Considering the gender earnings gap among workers, we divide the set of workers $\mathcal{D}$ into two subgroups: $\mathcal{D}^{(f)}$ for female workers, $\mathcal{D}^{(m)}$ for male workers. So far, we have defined the set of subgroups $\mathcal{S} := \{f, m\}$ and an element of $\mathcal{S}$ is denoted by $s$. Consequently, we have the average return rate for each subgroup: $\mu_t^{(s)} = \frac{1}{|\mathcal{D}^{(s)}|} \sum_{j \in \mathcal{D}^{(s)}} U_t^j / \Lambda_t^j$, for $s \in \mathcal{S}$.

An important property of the GE index allows one to decompose the overall inequality into the inequality within subgroups and the differences between subgroups [46, 47], such that

$$
GE(1)_t = \underbrace{\sum_{s \in \mathcal{S}} v^{(s)} \omega_t^{(s)} GE(1)_t^{(s)}}_{\text{within subgroup term}} + \underbrace{\sum_{s \in \mathcal{S}} v^{(s)} \omega_t^{(s)} \ln \omega_t^{(s)}}_{\text{between subgroup term}},
\tag{5}
$$

$$
GE(0)_t = \underbrace{\sum_{s \in \mathcal{S}} v^{(s)} GE(0)_t^{(s)}}_{\text{within subgroup term}} + \underbrace{\sum_{s \in \mathcal{S}} v^{(s)} \ln \dfrac{1}{\omega_t^{(s)}}}_{\text{between-subgroup term}},
\tag{6}
$$

where $v^{(s)} := \frac{|\mathcal{D}^{(s)}|}{|\mathcal{D}|}$ and $\omega_t^{(s)} := \frac{\mu_t^{(s)}}{\mu_t}$. Note that $\text{GE}(1)_t^{(s)}$, $\text{GE}(0)_t^{(s)}$ are GE(1) and GE(0) indices of subgroup $s$ in period $t$ but we will not use them in the following text. The between-subgroup terms of GE(1), GE(0) indices in Eqs (5) and (6) can give us some ideas for defining gender gap of Worker-Utility. We gather all the above ideas with a straightforward definition to be Inter-Fair$_t$ till period $t \in \mathcal{T}$ in (7).

$$\text{Inter-Fair}_t := \begin{cases} 1. \sum_{s \in \mathcal{S}} v^{(s)} \omega_t^{(s)} \ln \omega_t^{(s)} & \text{GE}(1)_t \\ 2. \sum_{s \in \mathcal{S}} v^{(s)} \ln \frac{1}{\omega_t^{(s)}} & \text{GE}(0)_t \\ 3. \left| \mu_t^{(f)} - \mu_t^{(m)} \right| \end{cases} \tag{7}$$

Please note that the formula of Inter-Fair$_t$ can easily be extended to the cases of multiple subgroups or overlapping subgroups of sensitive attributes.

## 4 Formulating Intra-, Inter-Fair and Customer-Care

So far, we have mentioned three objectives that we wish to maximise (Customer-Care) or minimise (Intra- and Inter-fairness). However, the three objectives are usually in conflict with each other. A natural approach to multi-objective problems utilises a convexification, e.g., a weighted sum of the objectives.

The resulting natural formulation entails selecting scalar weights $\gamma^{(1)}$, $\gamma^{(2)}$, $\gamma^{(3)}$ and minimising a convexified optimisation problem defined in formulation (8), in binary variables $M_t^{(i,j)}$, for $i \in \mathcal{C}_t, j \in \mathcal{D}$, and auxiliary variables $u_t^j$, for $j \in \mathcal{D}$. The input of formulation (8) in period $t$ is composed of $d_t^{(i,j)}$, for $i \in \mathcal{C}_t, j \in \mathcal{D}$; $A_t^j$, $\text{pay}_t^i$, for $i \in \mathcal{C}_t$; $U_{t-1}^j$, $\Lambda_t^j$, for $j \in \mathcal{D}$; $|\mathcal{D}^{(s)}|$, for $s \in \mathcal{S}$; and $\gamma^{(1)}$, $\gamma^{(2)}$, $\gamma^{(3)}$.

$$\min_{M_t^{(i,j)}, u_t^j} \gamma^{(1)} \, \text{Intra-Fair} + \gamma^{(2)} \, \text{Intra-Fair} - \gamma^{(3)} \, \text{Customer-Care}$$

$$\text{s.t.} \quad \sum_{i \in \mathcal{C}_t} M_t^{(i,j)} \left( \text{pay}_t^i - d_t^{(i,j)} \right) = u_t^j, \quad j \in \mathcal{D}^{A_t=1} \tag{8a}$$

$$0 = u_t^j, \quad j \in \mathcal{D}^{A_t=0} \tag{8b}$$

$$A_t^j - \sum_{i \in \mathcal{C}_t} M_t^{(i,j)} \geq 0, \quad j \in \mathcal{D} \tag{8c}$$

$$\sum_{j \in \mathcal{D}} M_t^{(i,j)} = 1, \quad i \in \mathcal{C}_t \tag{8d}$$

$$M_t^{(i,j)} \in \{0,1\}, \quad i \in \mathcal{C}_t, \; j \in \mathcal{D} \tag{8e}$$

Apparently, formulation (8) is a mixed-integer quadratic problem. In general, mixed-integer quadratic problems are NP-Hard to solve, but in this particular case, one can utilise an augmented Lagrangian relaxation that can be solved efficiently.

## 4.1 Hidden convexity

There has been much work on "hidden convexity" [48–50], i.e., non-convex optimization problems that are solvable in polynomial time, often by reformulating them as convex optimization problems. Let us recall two textbook examples of hidden convexity, before we show how to reformulate Formulation (8) such that the hidden convexity becomes apparent. First, consider the search for a shortest path in a digraph. In the most common formulation [51, Eq. (1–4)], variables $x_{ij}$ are restricted to binary values, making the the feasible set apparently non-convex. There is, however, convexity hidden in the fact that every basic optimal solution of the linear programming relaxation $0 \leq x_{ij} \leq 1$ (when one exists) has all variables equal to 0 or 1, and variables whose values are equal to 1 correspond to an a directed path from $s$ to $t$. Next, consider the example max $x$ s.t. $x^2 + y^2 = 1$ of [50]. The feasible set is a circle, which is non-convex, unlike the disc. At the same time, it is easy to see that the variable $y$ is redundant. We can reformulate the problem to max $x$ s.t. $0 \leq x \leq 1$ and it becomes clear the solution is $x = 1, y = 0$. This, in effect, "solves away" the non-convex constraint. Similar techniques [48, Section 2] can be applied to indefinite quadratically-constrained problems more broadly.

Returning to our setting, we can formulate the so-called Augmented Lagrangian relaxation of linear equality constraints in (8a, 8b, 8d, 8e) by considering their squared violations. For constraints (8a, 8b, 8d), their squared violations are straightforward to formulate. For instance, the violation of constraint (8b) is $(0 - u_t^j)^2, j \in \mathcal{D}^{A_t=0}$. As in [52], we can consider the violations squared in the objective with positive multipliers $\phi_j^{(1)}, \phi_j^{(2)}, \phi_i^{(3)}$ for the violation squared of constraints 8a, 8b and 8d, respectively. By making the multipliers larger, we penalise constraint violations more severely, thereby forcing the minimiser of the objective with quadratic penalty terms closer to the feasible region for the constrained problem. For integrality constraints $M_t^{(i,j)} \in \{0, 1\}$ in (8d), one may consider replacing it with a new constraint:

$$M_t^{(i,j)} \left( M_t^{(i,j)} - 1 \right) = 0. \tag{9}$$

and bring it to the objective with a multiplier $\phi_{i,j}^{(4)}$, while one can also limit $M_t^{(i,j)}$ to be non-negative. The resulting formulation:

$$
\begin{aligned}
\min_{M_t^{(i,j)}, u_t^j} \quad & \gamma^{(1)} \text{ Intra-Fair} + \gamma^{(2)} \text{ Intra-Fair} - \gamma^{(3)} \text{ Customer-Care} \\
& + \sum_{j \in \mathcal{D}^{A_t=1}} \phi_j^{(1)} \left( \sum_{i \in \mathcal{C}_t} M_t^{(i,j)} \left( \overset{i}{\text{pay}}_t - d_t^{(i,j)} \right) - u_t^j \right)^2 + \sum_{j \in \mathcal{D}^{A_t=0}} \phi_j^{(2)} (u_t^j)^2 \\
& + \sum_{i \in \mathcal{C}_t} \phi_i^{(3)} \left( \sum_{j \in \mathcal{D}} M_t^{(i,j)} - 1 \right)^2 + \sum_{i \in \mathcal{C}_t} \sum_{j \in \mathcal{D}} \phi_{i,j}^{(4)} \, M_t^{(i,j)} (M_t^{(i,j)} - 1) \\
\text{s.t.} \quad & A_t^j - \sum_{i \in \mathcal{C}_t} M_t^{(i,j)} \geq 0, \quad j \in \mathcal{D}; \\
& M_t^{(i,j)} \geq 0, \quad i \in \mathcal{C}_t, \quad j \in \mathcal{D};
\end{aligned}
\tag{10}
$$

utilises continuous variables $M_t^{(i,j)}$, for $i \in \mathcal{C}_t, j \in \mathcal{D}$ and $u_t^j$, for $j \in \mathcal{D}$ and for each period $t$, while the input of the formulation is the same as that of the natural formulation (8). This turns out to be rather an interesting, non-trivial example of hidden convexity.

In particular, one can optimise over the non-convex augmented Lagrangian relaxation (10) to any fixed precision in polynomial time. Formally, for any $\epsilon > 0$, one can approximate the solution of the non-convex non-linear optimisation problem in (10) to $\epsilon$ precision on the

Turing machine in time polynomial in the dimension. The proof follows from Theorem 2 of [52] and Theorems 5.1 and 5.2 of [53]. In particular, (10) plays the role of the equivalent problem $P_E$ in (10) and we can approximate the semidefinite programming (SDP) relaxation B3 of [52] to any fixed precision in polynomial time, following the reasoning of Remark 5.1 of [53]. Notice that without considering the details of the bit-complexity of the representation of the solution, one could apply standard convergence analyses of primal-dual interior point methods for SDP bound B3 of Theorem 2 of [52], instead of the results of [53]. One could also consider time-varying extensions [54, 55], although this is outside of the scope of the present draft.

## 5 Experiments

Experimentally, we set out to answer these questions:

Q1. What combination of the individual measures of Intra- and Inter-fairness in an objective function of the market-clearing mechanism works the best, with respect to the individual measures of Intra- and Inter-fairness?

Q2. What is the trade-off between the Intra- and Inter-fairness?

In the Section of Supporting information, we also address other questions, such as: Can the proposed market-clearing mechanism be implemented efficiently? How stable is the trade-off in cross-validation? To answer these questions, we have implemented:

- the augmented Lagrangian formulation (10) using the tssos library of Wang et al [56, 57]. For reference, we have also implemented the non-convex version using IBM CPLEX Optimizer Python API (CPLEX).

- the natural formulation (8) in two modelling frameworks: IBM CPLEX Optimizer Python API and IBM Decision Optimisation CPLEX Optimizer API for Python (DOcplex).

  Subsequently, we used:

- IBM CPLEX Optimizer as Mixed Integer Linear Programming (MILP) solver via its native API. This provides global optima of (8), but in the worst case, in exponential time.

- IBM CP Optimizer as a Constraint Programming (CP) solver via DOcplex. This does not provide guarantees of global optimality.

- SDPA as an SDP solver via the tssos library. This provides global optima of (10) in the worst case in polynomial time, as discussed in Section 4.1. As the degrees of objective and constraints in (10) are less than or equal to 2, we used the first order of the correlative and term sparsity exploiting moment/SOS (CS-TSSOS) hierarchy [57].

We include the comparison of runtime across the three solvers and two formulations in the Section of Supporting information, which shows that the augmented Lagrangian relaxation does improve the runtime, compared with the exponential-time algorithms of [11] for optimising Proportional Fairness.

We tested the methods on a subset of the well-known 2018 Yellow Taxi Trip Dataset under licence at terms-of-use.page, which were collected and provided to the NYC Taxi and Limousine Commission (TLC) by technology providers authorised under the Taxicab & Livery Passenger Enhancement Programs (TPEP/LPEP). This dataset includes fields capturing pick-up and drop-off dates/times, pick-up and drop-off locations, trip distances, itemised fares, rate types, payment types, and driver-reported passenger counts. While the dataset is large, the matching of jobs (rides) to workers (drivers) is typically run frequently, for small batches of jobs (rides) and workers (drivers). Indeed: making the batches larger would leave both

customers and workers to wait longer for their matches. Even during rush hours, it may be preferable to narrow down the search area (to a few closest drivers) and utilise the same small batch sizes. Therefore, we have run a repeated $k$-fold cross-validation with randomly chosen batches of jobs from the dataset. (See Section 5.2 and Section of Supporting information for more details).

To formalise each batch, suppose that there are 10 jobs in period $t$, i.e., $\mathcal{C}_t = \{i_0, \ldots, i_9\}$. We have 20 workers in the system in total, i.e., $\mathcal{D} = \{j_0, \ldots, j_{19}\}$, half of which $j_2, j_4, \ldots, j_{18}$ are females. Hence, $|\mathcal{D}| = 20$, $|\mathcal{D}^{(f)}| = |\mathcal{D}^{(m)}| = 10$. All workers are all available in period $t$, thus $A_t^j = 1$, for $j \in \mathcal{D}$. For simplicity, we assume in period $t$ all workers are new to the system, hence their accumulated utility $U_{t-1}^j$ and accumulated active time $\Lambda_{t-1}^j$ are zero. We define the maximal return rate til period $t-1$ as $m_{t-1} := \max_{k \in \mathcal{D}} U_{t-1}^k / \Lambda_{t-1}^k$. The values of $\text{pay}_t^i$, $i \in \mathcal{C}_t$ correspond to the trip distances within the batch of jobs. The values of $d_t^{(i,j)}$, $i \in \mathcal{C}_t, j \in \mathcal{D}$ are uniform random variables from the interval of $[0, 0.5]$. $\phi_j^{(1)}, \phi_i^{(2)}$ are both set to 10 in following experiments so that the optimum of augmented Lagrangian relaxation in (10) can be reached. We do not consider Customer-Care in experiments focusing on the trade-off between notions of fairness and set $\gamma^{(3)}$ to 0 in all formulations.

## 5.1 A comparison of the fairness notions (Q1)

To answer Q1, notice that we have three types of Inter-fairness in (7), which are called Inter 1,..., Inter 3, and five types of Intra-fairness in (4), which are called Intra 1, ..., Intra 5. We can optimise with respect to a combination of these and evaluate the Optimiser with respect to any of these fairness measures *post hoc*. To reduce the number of comparisons to be made, but to preserve comparability with the results of [11], who minimise Intra 5 without considering Inter-fairness, we use Intra 5 in the objective throughout, possibly combined with Inter 1, Inter 2 or Inter 3 with equal weights. In this way, we obtain four explicit formulations: Intra 5 + Inter 1, Intra 5 + Inter 2, Intra 5 + Inter 3, and "Sühr et al. 2019", where $\gamma^{(2)} = 0$ and $\gamma^{(1)} = 1$.

In Fig 1, we compare the four formulations, as suggested by four different colours, in terms of the mean values of Intra 2 ($\text{GE}(1)_t$), Intra 3 ($\text{GE}(0)_t$), Intra 4 ($\text{Gini}_t$), and Inter 1, Inter 2, Inter 3 for 30 runs, as suggested by the bars. The four formulations are implemented for 5 trials ($4 \times 5$ runs in total) in IBM® Decision Optimisation CPLEX Optimizer Modelling for Python (DOcplex), with the different batch of jobs randomly chosen from the dataset and different $d_t^{(i,j)}$ for each trial. Although a batch is to match 10 jobs to 20 workers throughout the paper, the experiments in Fig 1 use 5 jobs and 10 workers for each batch. Each experiment gives a matching of workers and jobs, from which we can calculate the values of Intra 2 ($\text{GE}(1)_t$), Intra 3 ($\text{GE}(0)_t$), Intra 4 ($\text{Gini}_t$), and Inter 1, Inter 2, Inter 3 for each implementation. The black vertical line on top of each bar suggests the mean ± one standard deviation.

In terms of Intra-fairness, we included Intra 5 in the objective throughout, and the results are comparable across all four formulations and all three measures of Intra-fairness used in the evaluation. In terms of Inter-fairness, however, we show that including the Inter-fairness in the objective, improves the Inter 1, Inter 2, Inter 3 measures of Inter-fairness, without a detrimental impact on the Intra-fairness. The minimisation of a combination of Intra- and Inter-fairness, especially Intra 5 + Inter 3, significantly decreases Inter-fairness, does not lower measures of Intra-fairness as "Sühr et al. 2019".

## 5.2 A trade-off between Intra- & Inter-Fair (Q2)

To answer Q2, notice that Intra-, Inter-Fair, and Customer-Care objectives are in conflict, but we can trace the set of Pareto optimal solutions. Pareto optimal solutions provide an optimal

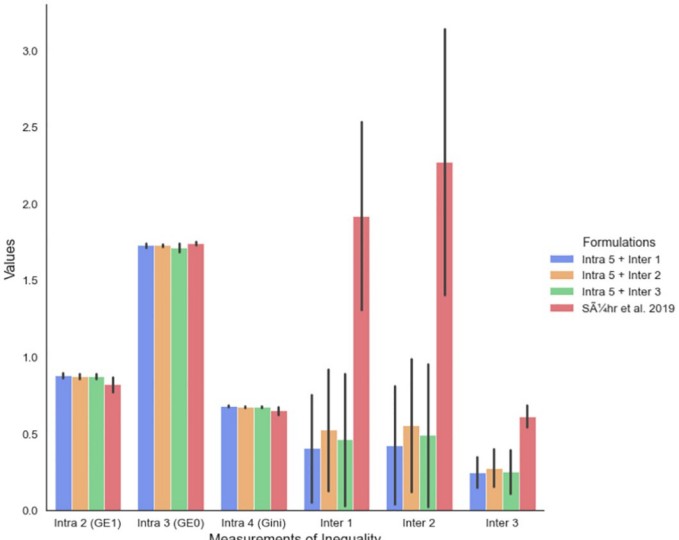

**Fig 1. Experimental results of four formulations implemented for 5 trials (4 × 5 runs in total), in DOcplex, with different batch of jobs from NYC taxi dataset and different $d_t^{(i,j)}$ for each trial.** The mean values of six indices for the 20 implementations are denoted as bars, while the black vertical line on top of each bar denotes the mean ± one standard deviation.

trade-off between multiple objectives, in the sense that it is not possible to improve one objective without detriment to another objectives. The set of all Pareto optimal solutions is known as the Pareto front.

Here, we focus on the trade-off between Intra- and Inter-Fair, while the same procedures can be extended to Customer-Care too (see [17, 58]). For the formulation "Intra 5 + Inter 1", the value of $\gamma^{(1)}$ is uniformly sampled from [0.5, 0.9] with the interval of 0.1 and $\gamma^{(2)}$ is set to be $1 - \gamma^{(1)}$. Note that the term "(L)" in the legend shows that we use the augmented Lagrangian formulation (10) instead of the natural formulation (8) for implementation. In "Sühr et al. 2019", we take $\gamma^{(1)} = 1$, $\gamma^{(2)} = 0$. We implement each formulation for 25 trials (25 × 2 runs in total), with a batch of jobs chosen randomly from the dataset being the same within a single trial.

In Fig 2, we present the results of each run using `tssos` with one dot. For each run, we calculate the values of Intra 2,. . .,Intra 4 and Inter 1,. . .,Inter 3 from the results *post hoc*. We use the results to position the dots in the corresponding subplots. The results of "Sühr et al. 2019 (L)" [11], which are shown in red dots, are overwhelmingly dominated by the results of our formulation Intra 5 + Inter 3 (L), which are shown by green dots. Correspondingly, Pareto front, which is shown by a black curve, is composed mostly of results of Intra 5 + Inter 3 (L). While the results should not be particularly surprising, it suggests that Inter-fairness can be included in the objective without any harm in terms of Intra-fairness.

**Robustness analysis.** Each batch contains 10 jobs and 20 workers. This would be enough for the matching of jobs (rides) to workers (drivers), which typically runs frequently for small batches. There might be concerns about i) whether different combinations of $(\gamma_1, \gamma_2)$ would result in irreconcilable performance and ii) whether the batch of jobs can be representative of the behaviour on the Yellow Taxi Trip Dataset. We have hence performed Intra 5 + Inter 3 (L) for 250 trials, where in each trial, a new batch of jobs is sampled from the Yellow Taxi Trip Dataset as described above, and the tuple $(\gamma_1, \gamma_2)$ for this formulation was uniformly chosen from the five combinations (0.5, 0.5), (0.6, 0.4), . . ., (0.9, 0.1). In Fig 3, we present every pair of

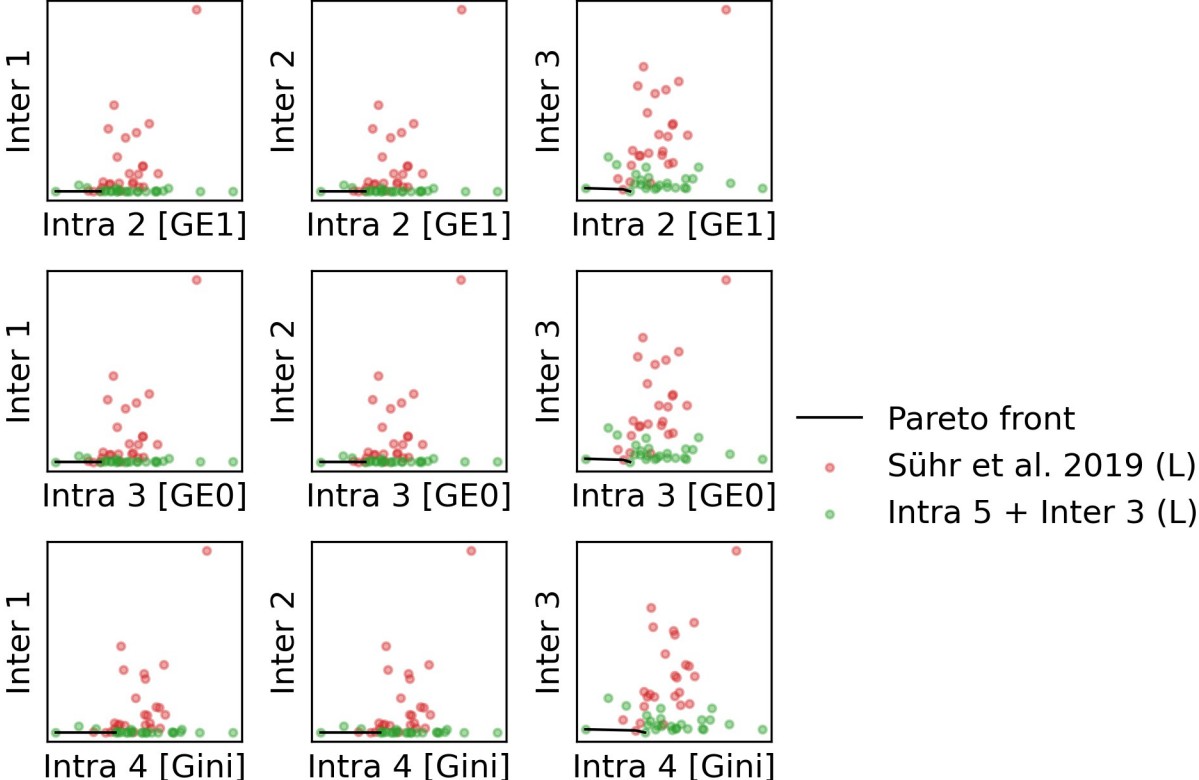

**Fig 2. An illustration of the trade-off between Intra- and Inter-fairness in a single trial.** The position of each dot represents the value of Intra- and Inter-fairness from one experiment using the implementation of the augmented Lagrangian formulation (10) in `tssos`. Red dots suggest the use of $\gamma^{(1)} = 1$, $\gamma^{(2)} = 0$, as in [11]. Green dots suggest the use of Intra 5 + Inter 3 (L). The Pareto front is shown by a black curve. See the Section of Supporting information for further details.

trade-off between Intra- and Inter-fairness, from the 250 trials, by a 3 × 3 plot. In each subplot, a dot represents the values of the corresponding Intra- and Inter-fairness of one trial. The histograms on the top and the right sides show the distribution of this pair of Intra- and Inter-fairness among the 250 trials. The concentration of all pairs of trade-offs implies the robustness of our formulation.

## 5.3 Runtime

In Fig 4, we compare the runtime of formulations of [11] and our method (Intra 5 + Inter 3) in CP Optimizer (CP), CPLEX (MILP), and `tssos` (SDP), with or without augmented Lagrangian relaxation, respectively. We use △ and ▽ markers to distinguish between the original formulation in (8) and its Lagrangian variant of (10) (suggested by "(L)"). We use green and red colours to differentiate between the formulation in [11] and Intra 5 + Inter 3. Solvers are separated by different shades of colours. The subplots on the right give the overview of the runtime of all formulations and all solvers, against the number of variables. Alongside the mean across 5 runs presented by a curve, there is a shaded error band at mean ± 1 standard deviations. The subplots on the left give a zoom-in effect of the right ones, without shaded error bands.

## 6 Conclusions

We have introduced the notion of Inter-fairness across subgroups in a two-sided market, in addition to Intra-subgroup fairness, among individuals within a subgroup. We have explored

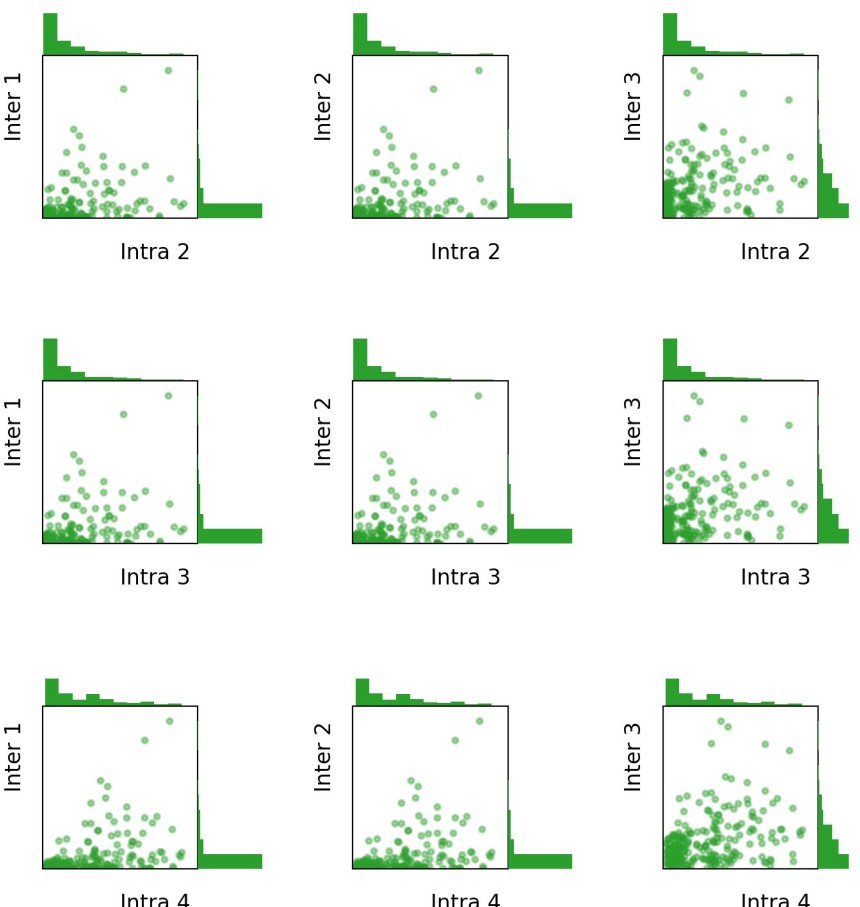

**Fig 3. All pairs of trade-offs between Intra- and Inter-fairness from 250 trials of formulation Intra 5 + Inter 3 (L).**
Each trial uses distinct batch of jobs and its parameters ($\gamma_1$, $\gamma_2$) uniformly vary from (0.5, 0.5) to (0.9, 0.1) with the
interval of 0.1. Each dot in a subplot represents the values of the corresponding Intra- and Inter-fairness measured *post
hoc* for the result of one trial. The histograms on the top and the right sides show the distribution of the Intra- and
Inter-fairness among the 250 trials, respectively.

the trade-off between Intra-group and Inter-group fairness on the example of a ride-hailing
using the Yellow Taxi Trip Dataset. Furthermore, we have shown that considering both objec-
tives at the same time is possible within the market-clearing, and leads to an efficiently approx-
imable non-convex augmented Lagrangian formulation. Our implementation is available
online at Fairness-in-Two-Sided-Markets. We have presented promising computational
results for several numerical routines; an approach considering a convex combination of
objectives Intra 5 and Inter 3 seemed particularly promising.

  We hope that the notion, the insights, and algorithms may be applicable across a range of
two-sided markets, such as online labour platforms and college admissions, and perhaps
extended to a comprehensive framework for multiple fairness criteria, such as the fairness res-
olution model in [24] and the permutation testing framework in [59]. One could consider
robust [18, 19] extensions or combining with localisation of Pareto-optimal equilibrium points
[60, 61]. One could also consider finite-horizon properties of the trajectory of solutions of (10)
using recent results [54] on time-varying semidefinite programming. Finally, one could also
ask numerous questions [62] in relation to the long-run behaviour of such systems, whose
answers may be based on recent work on the ergodic control of ensembles [63, 64]. Overall, we

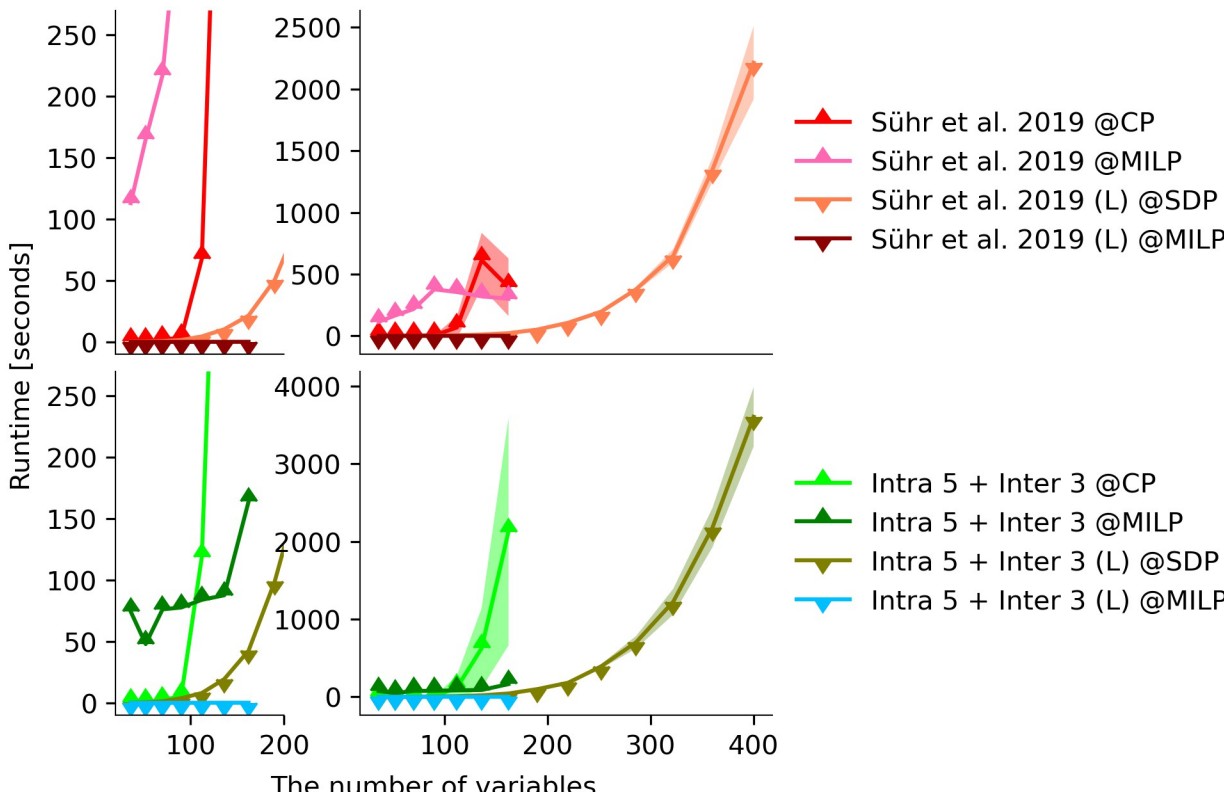

**Fig 4. The runtime of formulations "Sühr et al. 2019" and our method Intra 5 + Inter 3 using CP Optimizer (CP), CPLEX (MILP), and an SDP solver (SDPA), with or without augmented Lagrangian relaxation, respectively.** The four types of red curves denote the runtime of formulation "Sühr et al. 2019" (with △ markers) and its Lagrangian variant (with ▽ markers) against the number of variables. The four types of green curves denote that of formulations Intra 5 + Inter 3 and its variant. Subplots on the right present the mean runtime and mean ± one standard deviations across 5 runs by curves with shaded error bands. The subplots on the left give a zoom-in effect of the right ones, without shaded error bands.

imagine that this may contribute towards a theoretical foundation for the study of fairness in two-sided markets.

## Supporting information

**S1 Appendix. Experiments of cross validations.**
(PDF)

**S1 Fig. The trade-off between Intra- and Inter-fairness across 5 folds.** Fold 1 is presented in Fig 2. Each subplot presents trade-off between a pair of Intra- and Inter-fairness with Pareto-Front. Each green dot represents one trial of the implementation of augmented Lagrangian formulation (10) in tssos. Red dots denote a trial of "Sühr et al. 2019 (L)". The experimental details are in S1 Appendix.
(TIF)

**S2 Fig. The distributions of Intra- and Inter-fairness trade-off from 100 trials (folds) of formulation Intra 5 + Inter 3 (L) via tssos.** In each subplot, each dot represents the values of corresponding Intra-fairness and Inter-fairness notions of one trial. The histograms on top and on left side show the distribution of Intra-fairness and Inter-fairness of the 100 trials. The

experimental details are in S1 Appendix.
(TIF)

**S3 Fig. As an analogy to the 5-fold cross validation in S1 Fig, the same implementation is repeated with 10-fold.** We show the trade-off plots of 10 new trials.
(TIF)

## Author Contributions

**Conceptualization:** Quan Zhou, Jakub Mareček, Robert Shorten.

**Data curation:** Quan Zhou, Jakub Mareček.

**Formal analysis:** Quan Zhou, Jakub Mareček, Robert Shorten.

**Funding acquisition:** Robert Shorten.

**Methodology:** Quan Zhou, Jakub Mareček, Robert Shorten.

**Project administration:** Jakub Mareček, Robert Shorten.

**Resources:** Jakub Mareček, Robert Shorten.

**Software:** Jakub Mareček.

**Supervision:** Jakub Mareček, Robert Shorten.

**Writing – original draft:** Quan Zhou, Jakub Mareček.

**Writing – review & editing:** Quan Zhou, Jakub Mareček, Robert Shorten.

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
