## [Decision Letter · Decision Letter 0]

1 Aug 2022

PONE-D-22-04438Subgroup Fairness in Two-Sided MarketsPLOS ONE

Dear Dr. Zhou,

Thank you for submitting your manuscript to PLOS ONE. After careful consideration, we feel that it has merit but does not fully meet PLOS ONE’s publication criteria as it currently stands. Therefore, we invite you to submit a revised version of the manuscript that addresses the points raised during the review process.

We look forward to receiving your revised manuscript.

Kind regards,

Ashwani Kumar, Ph.D.

Academic Editor

PLOS ONE

Journal Requirements:

“Quan’s and Bob’s work has been supported by the Science Foundation Ireland under 379 Grant 16/IA/4610. Jakub acknowledges support of the OP RDE funded project 380 CZ.02.1.01/0.0/0.0/16 019/0000765 “Research Center for Informatics”.

Reviewers' comments:

Reviewer's Responses to Questions

**Comments to the Author**

1. Is the manuscript technically sound, and do the data support the conclusions?

Reviewer #1: Yes

Reviewer #2: Yes

2. Has the statistical analysis been performed appropriately and rigorously? 

Reviewer #1: Yes

Reviewer #2: Yes

3. Have the authors made all data underlying the findings in their manuscript fully available?

Reviewer #1: Yes

Reviewer #2: Yes

4. Is the manuscript presented in an intelligible fashion and written in standard English?

Reviewer #1: Yes

Reviewer #2: Yes

5. Review Comments to the Author

Reviewer #1: I have now completed the review of the manuscript titled "Subgroup Fairness in Two-Sided Markets". The manuscript have introduced the notion of Inter-fairness across subgroups in a two-sided market,

in addition to Intra-subgroup fairness, among individuals within a subgroup. I have some suggestions to further improve the quality of the manuscript.

1. The introduction section requires some relevant articles which used ML and DL for various applications [1-6], and also include some more market fairness applications using ML. Please add these in the related work section.

2. Please provide the computational complexity of all the models, see CDLSTM, SMOTEDNN, DNNBOT, PCCNN etc.

4. What is the future scope of the proposed research, authors have described the limitations in good way, I suggest that these can be the future scope of the work.

5. These days ML and AI are utilized to solve other applications which are based on several parameters, I suggest to make a small paragraph which discusses the role of AI and ML methods, authors can use some of the references provided in the comments 1 and 2.

6. Cross validation need to be clarified, why 5 is chosen, pl see [7].

References

1. Deep Learning Based Modeling of Groundwater Storage Change

2. SMOTEDNN: A Novel Model for Air Pollution Forecasting and AQI Classification

3. CDLSTM: A Novel Model for Climate Change Forecasting

4. CNN Based Automated Weed Detection System Using UAV Imagery

5. Assessment of trends of land surface vegetation distribution, snow cover and temperature over entire Himachal Pradesh using MODIS datasets

6. DNNBoT: Deep Neural Network-Based Botnet Detection and Classification

7. Efficiency of artificial neural networks for glacier ice-thickness estimation: A case study in western Himalaya, India

Reviewer #2: This paper formulates the objective of the market-clearing problem as a weighted sum of measures of subgroup fairness (Inter-fairness), fairness within each subgroup (Intra-fairness), as well as utility for the customers (Customer-Care). The authors use a non-convex augmented Lagrangian relaxation of the objective and show that it can be approximated to any precision in polynomial time with respect to the number of market participants. Finally, the efficacy and scalability of the approach are demonstrated and different trade-offs between the objective's terms are explored on a taxi trip dataset.

Strong points:

* The variant of the market-clearing problem addressed in this paper is of considerable importance for understanding and promoting fairness in two-sided markers given the recent rise of the gig economy.

* Despite the non-convexity of the formulated objective (due to the inclusion of non-linear terms), the authors show that such an objective can be optimized by approximating an augmented Lagrangian relaxation that considers the violations squared of the objective's linear equality constraints. Moreover, the optimization can be carried out to any fixed precision in polynomial time. This allows for efficient computation of the proposed market-clearing mechanism.

* Comprehensive experimentation has been conducted on the well-known 2018 NYC Yellow Taxi Trip Dataset, the results of which suggest that (1) the proposed approach is effective and scalable, and that (2) a measure of inter-fairness can be included in the objective function without compromising intra-fairness.

* The formulation of the inter-fairness measure is general enough in the sense that it can easily be extended to cases with multiple subgroups or overlapping subgroups of sensitive attributes.

* The paper is well organized and written in a quite clear and comprehensive manner. Also, the used notation is clear and consistent throughout the entire paper.

Weak points:

* At the beginning of Section 5, the authors claim that they used "SDPA as an SDP solver via the tssos library" that "provides global optima of Figure 1 in the worst case in polynomial time". Nevertheless, I am slightly confused by this claim as how can one guarantee that the global optima would be found considering that the original objective and its augmented Lagrangian relaxation are non-convex?

* The trade-off parameters \\gamma^{(1)},\\gamma^{(2)} and \\gamma^{(3)} are fixed throughout the experiments. I am wondering why the authors did not consider to automatically optimize the values of these parameters in addition to the other weights? Although the authors were able to essentially "grid-search" through various combinations of the trade-off parameters on the Yellow Taxi Trip dataset; using the same exploration strategy to determine the trade-offs on larger datasets would be quite challenging and computationally intensive. In their response, I would encourage the authors to elaborate a bit on the possibility of learning/optimizing the trade-off parameters automatically.

* There are also a few minor points that require attention:

- Page 11, line 292: Consider replacing "independent T test" with "independent t-test".

- Figure 1 is not essentially a figure as such but rather a reformulation of the optimization problem. I believe that it should be marked and referenced as an Equation rather than a Figure. Thus, I would suggest replacing Figure 1 with an equation label.

- In the discussion of the experiments, Figure 4 is referenced and discussed before discussing Figure 3. In that case, the order of appearance of the two figures should be reversed.

Overall, the paper is well developed and seems to be a good fit for PLOS One. Although there are some weak points that require certain revisions to be made, I believe that the strengths of this paper outweigh its weaknesses.

6. PLOS authors have the option to publish the peer review history of their article (what does this mean?). If published, this will include your full peer review and any attached files.

Reviewer #1: **Yes: **Mohd Anul Haq

Reviewer #2: No

---

## [Author Response · Author response to Decision Letter 0]

8 Sep 2022

Response to Editorial Team:

> Please ensure that your manuscript meets PLOS ONE’s style requirements, including those for file naming.

We have re-formatted the manuscript to meet PLOS ONE’s style requirements.

> please ensure that you provide the correct grant numbers for the awards you received for your study in the ‘Funding Information’ section.

Thank you for this comment. We have now corrected this issue.

> Please include your amended statements within your cover letter; we will change the online submission form on your behalf.

We have removed the acknowledgements section. Please add the funding statement as follows: The work of Quan Zhou and Rober Shorten has been supported by the Science Foundation Ireland under 379 Grant 16/IA/4610. Jakub Mareček acknowledges support of the OP RDE funded project 380 CZ.02.1.01/0.0/0.0/16 019/0000765 “Research Center for Informatics”.

Response to Reviewer #1:

> 1. The introduction section requires some relevant articles which used ML and DL for various applications [1-6], and also include some more market fairness applications using ML.

We have added a paragraph of the applications of AL and ML in related work. We have added material related to fairness rather than the paper on environmental modelling. Please see blue texts in Section of Related work. We thank the reviewer for this comment.

> 2. Please provide the computational complexity of all the models, see CDLSTM, SMOTEDNN, DNNBOT, PCCNN etc.

While the models of CDLSTM, SMOTEDNN, DNNBOT, and PCCNN do not consider fairness, the best we could do is to explain that these are models without any fairness considerations, wherein can obtain critical points of unclear value using first-order methods. We imagine that it may be preferable to maintain silence and decorum.

> 4. What is the future scope of the proposed research, authors have described the limitations in good way, I suggest that these can be the future scope of the work.

We have added a paragraph suggesting ideas for future work. Please see blue texts in Section of Conclusion. We thank the reviewer.

> 5. These days ML and AI are utilized to solve other applications which are based on several parameters, I suggest to make a small paragraph which discusses the role of AI and ML methods, authors can use some of the references provided in the comments 1 and 2.

We have added a paragraph on applications of AL and ML in related work. This paragraph directs the reader to background work on applications of AI. Please see blue texts in Section of Related work.

> 6. Cross validation need to be clarified, why 5 is chosen, pl see [7].

5-fold cross validation is typical of that used in industry. 

Indeed, many books and standard references suggest the use of 5-fold validation. For example: (James et al. 2013, p. 184) have explained: "to summarize, there is a bias-variance trade-off associated with the choice of k in k-fold cross-validation. Typically, given these considerations, one performs k-fold cross-validation using k = 5 or k = 10, as these values have been shown empirically to yield test error rate estimates that suffer neither from excessively high bias nor from very high variance."

As stated by (Kuhn and Johnson 2013, p. 70), "the choice of k is usually 5 or 10, but there is no formal rule. As k gets larger, the difference in size between the training set and the resampling subsets gets smaller. As this difference decreases, the bias of the technique becomes smaller."

Accordingly, k-fold cross validation is usually performed with k = 5 or k = 10. We have chosen the former one. These references have been added to the manuscript.

Response to Reviewer #2:

> Weak points: * At the beginning of Section 5, the authors claim that they used “SDPA as an SDP solver via the tssos library” that “provides global optima of Figure 1 in the worst case in polynomial time”. 

> Nevertheless, I am slightly confused by this claim as how can one guarantee that the global optima would be found considering that the original objective and its augmented Lagrangian relaxation are non-convex?

The reviewer may recall the concept of “hidden convexity”, i.e., that there are non-convex optimization problems that are solvable in P (polynomial time). Let us recall a textbook exampleof hidden convexity.

• Consider, as another example, max x s.t. x2 + y2 = 1. The feasible set is a circle, which is non-convex (unlike the disc, which is). At the same time, it is easy to see that the variable y is redundant. We can simplify the problem to max x s.t. 0 ≤ x ≤ 1 and it becomes clear the solution is x = 1, y = 0.

Returning to our setting:

• The original non-convex problem (8) may or may not be NP-Hard. We make no claim on its complexity or the computability of global optima in P.

• However, its non-convex relaxation (10) is equivalent to a linear semidefinite program, as shown by S. Poljak, F. Rendl, and H. Wolkowicz (“A recipe for semidefinite relaxation for (0, 1)-quadratic programming,” Journal of Global Optimization, vol. 7, no. 1, pp. 51–73, 1995). We can hence claim to provide, up to any precision, an approximation to the global optimum of the non-convex relaxation (10), in polynomial time, by solving the linear semidefinite program.

We believe that this is rather a nice, non-trivial example of hidden convexity.

> * The trade-off parameters γ(1), γ(2) and γ(3) are fixed throughout the experiments. I am wondering why the authors did not consider to automatically optimize the values of these parameters in addition to the other weights?

> Although the authors were able to essentially “grid-search” through various combinations of the trade-off parameters on the Yellow Taxi Trip dataset; using the same exploration strategy to determine the trade-offs on larger datasets would be quite challenging and computationally intensive. 

> In their response, I would encourage the authors to elaborate a bit on the possibility of learning/optimizing the trade-off parameters automatically.

Thank you for your suggestion. This could easily have been incorporated as part of a sensitivity/robustness analysis. 

However, while important, we felt the core message of the paper is one of fairness, and the fairness metrics, and that a sensitivity analysis of the overall system design is a subject best discussed in detail in the context of a specific use-case. 

Consequently, we have included it in our ideas for future work. Please see blue texts in Section of Conclusion. We hope that this is acceptable.

> * There are also a few minor points that require attention: - Page 11, line 292: Consider replacing “independent T test” with “independent t-test”.

We have replaced all “independent T test” with “independent t-test”.

> - Figure 1 is not essentially a figure as such but rather a reformulation of the optimization problem. 

> I believe that it should be marked and referenced as an Equation rather than a Figure. Thus, I would suggest replacing Figure 1 with an equation label.

We have marked Figure 1 as an equation.

> - In the discussion of the experiments, Figure 4 is referenced and discussed before discussing Figure 3. In that case, the order of appearance of the two figures should be reversed.

We have moved Figure 4 forwards.

---

## [Decision Letter · Decision Letter 1]

3 Oct 2022

PONE-D-22-04438R1Subgroup Fairness in Two-Sided MarketsPLOS ONE

Dear Dr. Zhou,

Thank you for submitting your manuscript to PLOS ONE. After careful consideration, we feel that it has merit but does not fully meet PLOS ONE’s publication criteria as it currently stands. Therefore, we invite you to submit a revised version of the manuscript that addresses the points raised during the review process.

We look forward to receiving your revised manuscript.

Kind regards,

Ashwani Kumar, Ph.D.

Academic Editor

PLOS ONE

Reviewers' comments:

Reviewer's Responses to Questions

**Comments to the Author**

1. If the authors have adequately addressed your comments raised in a previous round of review and you feel that this manuscript is now acceptable for publication, you may indicate that here to bypass the “Comments to the Author” section, enter your conflict of interest statement in the “Confidential to Editor” section, and submit your "Accept" recommendation.

Reviewer #1: (No Response)

Reviewer #2: All comments have been addressed

Reviewer #3: (No Response)

2. Is the manuscript technically sound, and do the data support the conclusions?

Reviewer #1: Partly

Reviewer #2: Yes

Reviewer #3: Partly

3. Has the statistical analysis been performed appropriately and rigorously? 

Reviewer #1: No

Reviewer #2: Yes

Reviewer #3: N/A

4. Have the authors made all data underlying the findings in their manuscript fully available?

Reviewer #1: No

Reviewer #2: Yes

Reviewer #3: Yes

5. Is the manuscript presented in an intelligible fashion and written in standard English?

Reviewer #1: Yes

Reviewer #2: Yes

Reviewer #3: Yes

6. Review Comments to the Author

Reviewer #1: Dear Authors

I observed that the authors have their own perspective around fairness only. I am also surprised to see the response to the comments which again shows the narrow context drawn by authors to improve the clarity on validation, which is highly important for studies which used computation in any aspect or application. The response such as, "We imagine that it may be preferable to maintain silence and decorum" and "the choice of k is usually 5 or 10, shown very limited perspective.

Reviewer #2: The authors have addressed the weak points suggested in my previous review, provided a point-by-point response to each and revised the manuscript accordingly. Therefore, I would suggest that this revised version is considered for publication in PLOS One if the other reviewers share the same view.

Reviewer #3: The research content of this manuscript is substantial. "Subgroup Fairness in Two-Sided Markets" is a research topic with economic-value and social-value.

The manuscript provided many descriptions and analyses of the

experimental result. However, this method or algorithm lacks robustness analysis and generalization analysis.

In addition, the experimental data set seems to be a little small, so it is recommended to replace it with a larger data set.

7. PLOS authors have the option to publish the peer review history of their article (what does this mean?). If published, this will include your full peer review and any attached files.

Reviewer #1: No

Reviewer #2: No

Reviewer #3: No

---

## [Author Response · Author response to Decision Letter 1]

22 Nov 2022

Response to Reviewer 1: 

> Dear Authors I observed that the authors have their own perspective around fairness only. I am also surprised to see the response to the comments which again shows the narrow context drawn by authors to improve the clarity on validation, which is highly important for studies which used computation in any aspect or application. 

We have extended our experiments in cross validation and robustness analysis in Section 5 and Supporting information.

> The response such as [...] "the choice of k is usually 5 or 10, shown very limited perspective.

As we have explained earlier, 5-fold cross validation is typically used in industry. Indeed, many books and standard references suggest the use of 5-fold validation. For example: (James et al. 2013, p. 184) have explained:

"To summarize, there is a bias-variance trade-off associated with the choice of k in k-fold cross-validation. Typically, given these considerations, one performs k-fold cross-validation using k = 5 or k = 10, as these values have been shown empirically to yield test error rate estimates that suffer neither from excessively high bias nor from very high variance."

As stated by (Kuhn and Johnson 2013, p. 70), 

"The choice of k is usually 5 or 10, but there is no formal rule. As k gets larger, the difference in size between the training set and the resampling subsets gets smaller. As this difference decreases, the bias of the technique becomes smaller."

These references have been added to the manuscript. 

On the other hand, we have conducted both k = 5 (Fig 2 \\& S1 Fig) and k = 10 (S3 Fig).

Response to Reviewer 2: 

> The authors have addressed the weak points suggested in my previous review, provided a point-by-point response to each and revised the manuscript accordingly. Therefore, I would suggest that this revised version is considered for publication in PLOS One if the other reviewers share the same view.

Thank you!

Response to Reviewer 3:

> The research content of this manuscript is substantial. "Subgroup Fairness in Two-Sided Markets" is a research topic with economic-value and social-value.

Thank you!

> The manuscript provided many descriptions and analyses of the experimental result. However, this method or algorithm lacks robustness analysis and generalization analysis.

We have added a discussion of robustness analysis in the end of Section 5.2, and added a new experiment accordingly. We hope that this addresses the concern of the reviewer.

> In addition, the experimental data set seems to be a little small, so it is recommended to replace it with a larger data set.

We have replaced all experiments with experiments on a larger dataset. We have also extended the batch sizes used for runtime comparison in Fig 4.

---

## [Decision Letter · Decision Letter 2]

24 Jan 2023

Subgroup Fairness in Two-Sided Markets

PONE-D-22-04438R2

Dear Dr. Zhou,

We’re pleased to inform you that your manuscript has been judged scientifically suitable for publication and will be formally accepted for publication once it meets all outstanding technical requirements.

Kind regards,

Ashwani Kumar, Ph.D.

Academic Editor

PLOS ONE

Additional Editor Comments (optional):

Reviewers' comments:

Reviewer's Responses to Questions

**Comments to the Author**

1. If the authors have adequately addressed your comments raised in a previous round of review and you feel that this manuscript is now acceptable for publication, you may indicate that here to bypass the “Comments to the Author” section, enter your conflict of interest statement in the “Confidential to Editor” section, and submit your "Accept" recommendation.

Reviewer #1: (No Response)

Reviewer #2: All comments have been addressed

Reviewer #3: All comments have been addressed

2. Is the manuscript technically sound, and do the data support the conclusions?

Reviewer #1: No

Reviewer #2: Yes

Reviewer #3: Yes

3. Has the statistical analysis been performed appropriately and rigorously? 

Reviewer #1: No

Reviewer #2: Yes

Reviewer #3: Yes

4. Have the authors made all data underlying the findings in their manuscript fully available?

Reviewer #1: Yes

Reviewer #2: Yes

Reviewer #3: Yes

5. Is the manuscript presented in an intelligible fashion and written in standard English?

Reviewer #1: Yes

Reviewer #2: Yes

Reviewer #3: Yes

6. Review Comments to the Author

Reviewer #1: Dear Authors

The technical suggestions tried to be managed somehow, I am not convinced with the revision which is general rather than technical.

Thank you so much.

Reviewer #2: The authors have already addressed all my comments in the initial review round, provided a point-by-point response to each and revised the manuscript accordingly. Therefore, I would suggest that this revised version is considered for publication in PLOS One if the other reviewers share the same view.

Reviewer #3: (No Response)

7. PLOS authors have the option to publish the peer review history of their article (what does this mean?). If published, this will include your full peer review and any attached files.

Reviewer #1: No

Reviewer #2: No

Reviewer #3: No

---

## [Editor Report · Acceptance letter]

31 Jan 2023

PONE-D-22-04438R2 

Subgroup fairness in two-sided markets 

Dear Dr. Zhou:

I'm pleased to inform you that your manuscript has been deemed suitable for publication in PLOS ONE. Congratulations! Your manuscript is now with our production department. 

Kind regards, 

on behalf of

Dr. Ashwani Kumar 

Academic Editor

PLOS ONE